# Widespread losses of pollinating insects in Britain

Gary D. Powney [1], Claire Carvell[1], Mike Edwards[2], Roger K. A. Morris[3], Helen E. Roy [1], Ben A. Woodcock [1] & Nick J. B. Isaac [1]

Pollination is a critical ecosystem service underpinning the productivity of agricultural systems across the world. Wild insect populations provide a substantial contribution to the productivity of many crops and seed set of wild flowers. However, large-scale evidence on species-specific trends among wild pollinators are lacking. Here we show substantial inter-specific variation in pollinator trends, based on occupancy models for 353 wild bee and hoverfly species in Great Britain between 1980 and 2013. Furthermore, we estimate a net loss of over 2.7 million occupied 1 km$^2$ grid cells across all species. Declines in pollinator evenness suggest that losses were concentrated in rare species. In addition, losses linked to specific habitats were identified, with a 55% decline among species associated with uplands. This contrasts with dominant crop pollinators, which increased by 12%, potentially in response agri-environment measures. The general declines highlight a fundamental deterioration in both wider biodiversity and non-crop pollination services.

[1] Biodiversity Science Area, Centre for Ecology and Hydrology, Wallingford OX10 8BB, UK. [2] BWARS (Bees, Wasps and Ants Recording Society), Leaside, Carron Lane, West Sussex GU29 9LB, UK. [3] UK Hoverfly Recording Scheme, Vine Street, Stamford, Lincolnshire PE9 1QE, UK. Correspondence and requests for materials should be addressed to G.D.P. (email: gary.powney@ceh.ac.uk)

Insect pollinators are vital for the maintenance of ecosystem health and for global food security, with 75% of crop species, 35% of global crop production, and up to 88% of flowering plant species[1] being dependent on insect pollinators to some extent[2,3]. However, substantial concern exists over their current and future conservation status[2,4,5]. Key threats to pollinators include agricultural intensification (particularly habitat loss and pesticide use), climate change and the spread of alien species[2,6,7]. Despite their importance, there is a critical absence of robust large-scale, species-specific estimates of distribution change for pollinating insects, in particular bees and hoverflies, which are considered some of the most important pollinators[4,8]. Published data on species-specific trends are currently only available from field-scale experiments typically spanning short time periods (<5 years) and spatially restricted to a limited number of sites[9]. Evidence at the large-scale comes from trends in aggregate metrics such as species richness and turnover[6,10–12]. Although useful, such metrics are insufficiently sensitive to identify pollination deficits nor are they suitable for developing International Strategic Goals (e.g., the Aichi Targets from the Convention on Biological Diversity, http://www.cbd.int/sp/targets/). In addition, given that pollination effectiveness and vulnerability to anthropogenic drivers differs between species[13,14], data on species-level trends are essential to understand the impacts of environmental change and the efficacy of conservation actions.

Biological records, defined as a record of a species at a given time and place, are a valuable but under-utilized source of data for estimating species trends[15]. The vast volume of these records, especially in western Europe and in particular Britain[15], allows the estimation of national-scale species-specific trend metrics spanning multiple decades. However, as biological records tend to be collected by large networks of volunteer recorders, they lack a standardized protocol and thus contain sampling bias. Considerable statistical issues need to be overcome if they are to be used for detecting genuine signals of change[16,17].

Here we take advantage of recent analytical developments to construct hierarchical Bayesian occupancy detection models[17–19] for 353 hoverfly and bee species, based on 715,392 biological records collected by the UK Hoverfly Recording Scheme (http://www.hoverfly.org.uk/) and the Bees, Wasps and Ants Recording Society (http://www.bwars.com/). We use these models to estimate national-scale species-level trends for Great Britain between 1980 and 2013. Our models estimate the proportion of occupied 1 km grid squares (henceforth occupancy) each year and are designed to account for incomplete and biased sampling in the raw data[17].

## Results

**Overall trends in pollinators.** We found widespread variation in the trends of wild pollinators in Britain, with individual species experiencing a range of trajectories between 1980 and 2013 (Fig. 1 and Supplementary Figure 1). Species-level trends, calculated as the annual growth rate in occupancy (percent change per year between the first and last year), reveal that a third of wild pollinator species (33%) have decreased over this period, approximately a tenth have increased, with the remaining species showing no clear trend (Supplementary Table 1 and Supplementary Figure 1). The balance of decreasing and increasing species was similar between bees and hoverflies (Supplementary Table 1). The direction and magnitude of the species-specific trend estimates, equate to a loss (net change) of 11 pollinator species (4 bees and 7 hoverflies) per 1 km grid cell between 1980 and 2013. Extrapolating these patterns to the whole of Great Britain (~240,000 1 km grid cells), our results estimate a net loss of over 2.7 million occupied 1 km grid cells for pollinator species between 1980 and 2013 (net change in the number of unique species by occupied 1 km grid cells). The magnitude of these changes highlights significant risks not just for regional pollinator communities, but also for the net provision of pollination services[20,21].

**Patterns of change among pollinator assemblages.** Contribution to pollination service is known to vary between species according

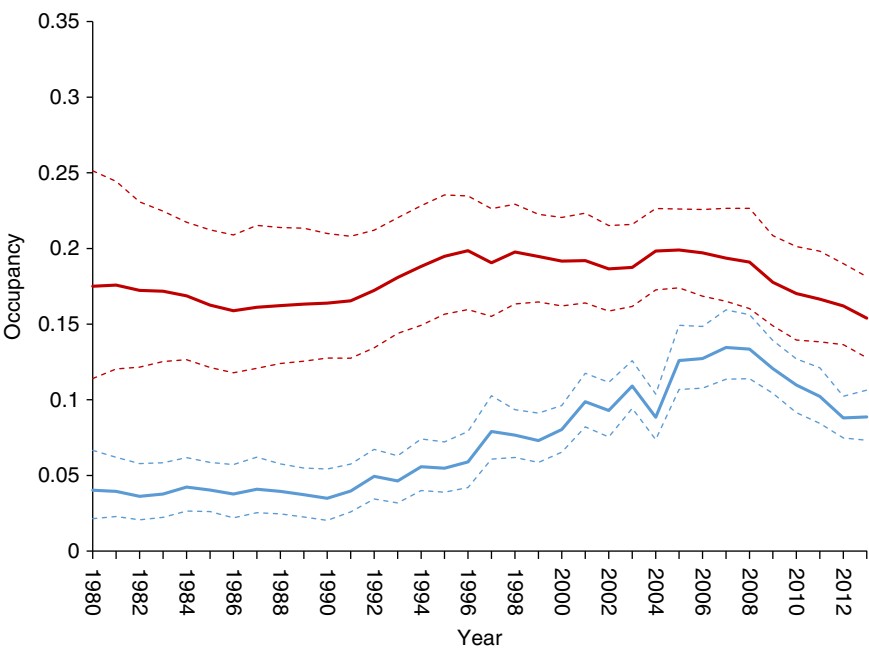

**Fig. 1** Trends of two example bee species illustrate contrasting patterns of change among species. Time series for *Bombus humilis* (blue) and *Colletes succinctus* (red) show the mean (solid line) and limits of the 95% credible intervals (dashed lines) of the posterior distribution of annual occupancy estimates

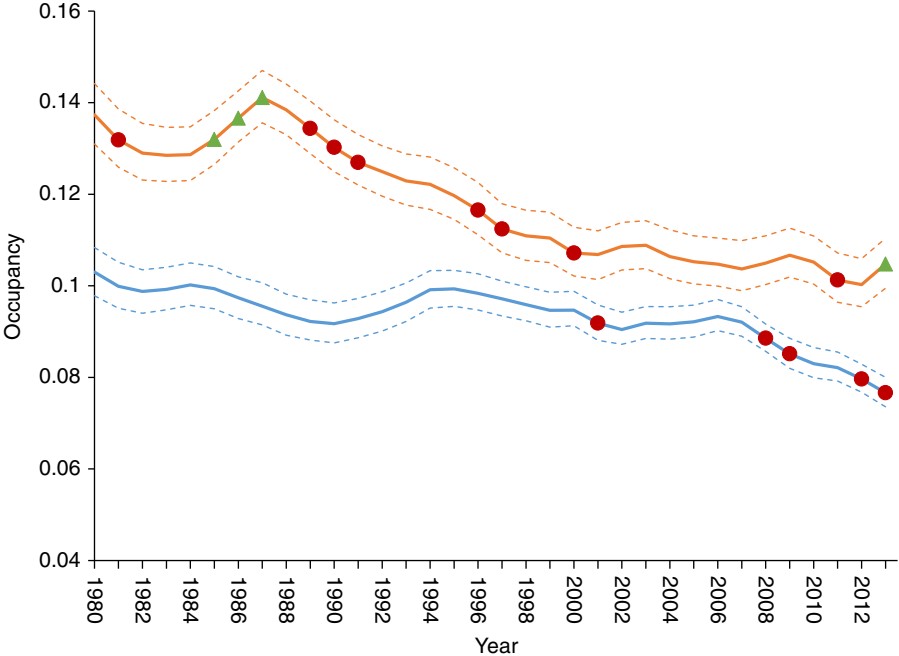

**Fig. 2** Contrasting patterns of change among major groups of pollinating insects. Trend lines show average occupancy of 1 km grid cells in Britain across all modelled bee (n = 139, blue) and hoverfly (n = 214, orange) species. Uncertainty is represented by the 95% credible intervals (delimited by dashed lines). Red circles and green triangles highlight years with notable decreases or increases, respectively. Notable years were defined as those where the upper (decreasing) or lower (increasing) 95% credible interval for the first derivative of occupancy did not span zero (see Supplementary Figure 2)

to their life history and ecological characteristics[22,23]. We therefore assessed long-term changes in mean occupancy for various trait-based subsets of pollinating insects. We found similar overall declines for bees (25% decline; 95% credible interval (CI): 21% to 30% decline; n = 139 species) and hoverflies (24% decline; 95% CI: 20% to 28% decline; n = 214), although there are marked differences between these two groups in the temporal pattern of declines (Fig. 2 and Supplementary Figure 2). Virtually all severe declines observed for overall bee occupancy occurred post 2007. By contrast, hoverflies declined steadily from 1987 to 2012. There are several key functional and ecological differences between bees and hoverflies, which could explain this pattern. Notably, most bees are fixed-place foragers whose early life stages are sheltered and actively provisioned by adults, whereas hoverflies move freely across landscapes and have juvenile stages filling a range of niches (e.g., aphidophagous, phytophagous, and detritivore) that are not directly cared for by adults. Understanding the contribution of these factors and their interaction with environmental change in explaining the contrasting trends of bees and hoverflies should be a priority for future research. Although most bee species declined, this was not the case for the subset of species identified as being key pollinators of a range of economically important European crops[14] (Supplementary Figure 3). On average, occupancy increased for these dominant crop pollinators by 12% (95% CI: 1% to 23%) from 1980 to 2013. In addition, we found notable changes in the eusocial bee species (including the bumblebees) (Supplementary Figure 4), for whom average occupancy increased by 38% (95% CI: 20% to 58%) compared with a decline of 32% (95% CI: 27% to 36% decline) for solitary bees (bees classified as non-eusocial in Supplementary Data 1). These increasing trends may be attributed to the widespread implementation of agri-environmental schemes specifically designed to support bumblebees in arable farming systems[24]. Furthermore, we found striking differences according to the species' geographic distributions. In particular, upland species showed declines of 55% (95% CI: 47% to 62% decline), whereas the average decline among

southern species was 25% (19% to 30% decline), with the majority of this change occurring since 2006 (Supplementary Figure 5). The apparent vulnerability of upland species may reflect retractions of the trailing (southern) range edges in response to climatic warming[25].

To further understand changes in pollinator assemblages, we used Simpson's evenness metric to assess the extent to which communities become dominated by a small number of widespread species[26]. We found little temporal variation in hoverfly evenness, but bees showed strong declines in evenness in the late 2000s (Fig. 3). The decline in bee evenness parallels the decline in mean occupancy of bees, suggesting losses in the late 2000s were concentrated among species with already small distributions. This result raises concerns around the fate of pollination services to wild flowers, given that more diverse communities are more effective in pollinating a wide range of wild flowers[2].

**Discussion**

Our findings fill an important gap in the evidence base on the status of wild pollinators. By providing species-level, national-scale estimates of change, our study found evidence of declines across a large proportion of pollinator species in Britain between 1980 and 2013. These overall declines are in addition to the losses that occurred before 1980, noted in previous studies[10], and are likely driven by a host of pressures known to act upon pollinators, including habitat loss, climate change, and pesticides[2,6,7,27]. In terms of conservation, it appears that current investment in agri-environmental schemes may have been effective in promoting pollinator populations on farmland, especially among the widespread common species responsible for crop pollination. However, as yields of pollinator-dependent crops are related to abundance as well as diversity of pollinators[28], the lack of standardized monitoring data limits our understanding of the link between change in species occupancy, local abundance, and in turn pollination deficit[28]. Although current conservation efforts

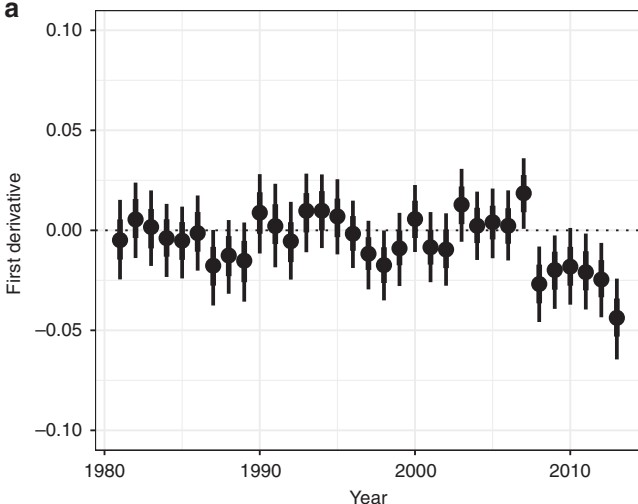

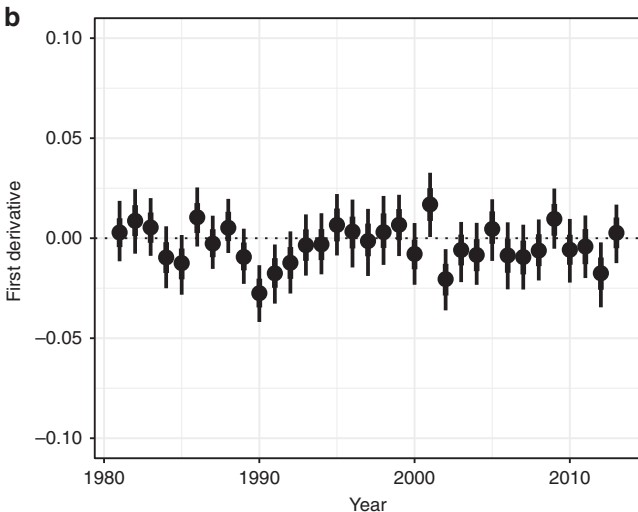

**Fig. 3** Annual estimates of change in assemblage evenness (first derivative of evenness). **a** Bee and **b** hoverfly assemblages. Points represent the median estimate of the posterior, with uncertainty presented as the limits of the credible intervals (thin = 95% CI, thick = 80% CI)

may have supported those crop pollinators, further effort is needed to develop new management approaches that restore habitat and food resources for pollinators across the wider landscape[29,30].

## Methods

**Distribution data.** Trends were estimated from occurrence records of hoverflies and bees extracted from the Hoverfly Recording Scheme (http://www.hoverfly.org.uk/) and the Bees, Wasps and Ants Recording Society (BWARS: http://www.bwars.com/). Combined, the dataset used in this study consisted of 715,392 (Hoverfly = 417,856, Bee = 297,536) records, defined as a unique combination of 1 km grid cell, date, and species. By excluding records pre-1980 and post-2013, we focussed on a core period of recording activity for both taxonomic groups. We excluded grid cells with < 2 years of data, removing the most poorly sampled regions. These observations constitute presence-only data, so we inferred non-detections from records of other species within the taxonomic group on the same grid cell and date (henceforth visit)[17,18]. The analysis was based on 12,849 and 12,076 unique 1 km grid cells for hoverflies and bees, respectively. The 1 km grid was chosen to reflect the scale at which hoverfly and bee populations use the landscape. Species with taxonomic issues during the time frame of the study and species not considered to be pollinators (following expert guidance from BWARS) were excluded from the analysis. In addition, we follow the species exclusion criteria of ref. [31], dropping species with fewer than 50 records. The final dataset was based on 139 bee and 214 hoverfly species (covering ~75% of the British bee and hoverfly fauna).

**Statistical analysis.** Much of these data were collected by volunteer recorders without specific sampling design. Therefore, the data contain a variety of forms of bias that inhibit the ability to extract robust trends from them. For example, the occurrence data suffered from temporal bias, with greater numbers of records in recent years. A host of techniques have been proposed to account for such bias while estimating trends, with recent studies suggesting hierarchical occupancy, models fitted within a Bayesian framework perform particularly well[17,18]. In this study, we used a Bayesian occupancy modelling approach based on the models of refs. [17] and [31], to estimate occupancy (the proportion of occupied 1 km grid cells) each year between 1980 and 2013 for each species. By using two hierarchically coupled sub-models (1 and 2, below), the occupancy model simultaneously estimates and accounts for variation in detectability, while estimating species presence for a given site, year combination.

$$\text{State model}: z_{it} \sim 2\text{Bernoulli}(\psi_{it}); \text{logit}(\psi_{it}) = b_t + u_i \quad (1)$$

$$\text{Observation model}: y_{itv}|z_{it} \sim \text{Bernoulli}(z_{it} * p_{itv}); \text{logit}(p_{itv}) = a_t + \delta_1.\text{DT2}_{itv} + \delta_2.\text{DT3}_{itv} \quad (2)$$

where, $z_{it}$ and $\psi_{it}$ are the true (unknown) occupancy and probability of occupancy of site $i$ in year $t$, respectively. $b_t$ and $u_i$ are categorical fixed and random effects for year and site (1 km grid cell), respectively. $Y_{itv}$ represents the observed data, this is a 1 or 0 based on whether the species was detected or not at site $i$, in year $t$, on visit $v$. $p_{itv}$ is the probability of detection at site $i$, in year $t$ on visit $v$, and is conditional upon $z_{it} = 1$. Probability of detection was modelled as a function of $a_t$ a random year level effect (accounting for variation in detectability over time), and $\delta_1$ and $\delta_2$ the effects of list categories 2 and 3, relative to category 1. For most species, we expect detectability to be lower on shorter lists, we therefore included list category ($\delta$) as a covariate in the detection model to account for variation in recorder effort. Visits were grouped into one of three categories based on the number of species recorded as follows: (1) single species lists, (2) short-day lists, 2 or 3 species recorded (DT2), and (3) comprehensive day lists, visits with >3 species recorded (DT3)[18]. Visits were defined separately for each taxonomic group; e.g., for any given bee occupancy model, the list length data was based solely on bee records.

Predicted presences ($z_{it}$) were combined to calculate the annual proportion of occupied sites (occupancy). For clarity, an occupancy value of 1 indicates the species occupied every 1 km grid cell included in the study (12,849 and 12,076 cells for hoverflies and bees, respectively). We used the random walk half-cauchy prior formulation of ref. [31], which enabled the sharing of information between the current and previous year in the state model, essentially adding a smoother for the annual occupancy estimates. We used uninformative priors for the remaining parameters within the model. For further detail of the occupancy model, see ref. [31]. Occupancy models were fitted using *R2jags*[32], with 3 chains, 20,000 iterations, and a burn in of 10,000 and a thinning rate of 3. This was sufficient to achieve convergence (Rhat < 1.1) for the vast majority of occupancy estimates across species and years: we retained the small minority of combinations for which Rhat > 1.1, as they are unlikely to exert directional bias on our high-level summary statistics.

As with all modelling approaches, the approach we used has several key assumptions. First, the model assumes no false presences, which we feel was a reasonable assumption given the data were validated by recording scheme organizers along with several automated checks. A second key assumption is that the detection sub-model reflects a true representation of observation process. There may be examples where this assumption is not met. For example, intense targeted surveys for certain species may not be fully accounted for in the detection model, leading to unreliable occupancy estimates for the species in question. Furthermore, strong temporal bias in recording intensity can lead to increased uncertainty in the occupancy estimates in earlier years. Bearing these issues and assumptions in mind, we chose hierarchical occupancy models, as they have been shown to perform well at dealing with such forms of bias[17], and although the detection model may not be perfect for all species, it is likely to be better than a model that ignores variation in detectability. It is worth noting that alongside these trends, the recent development of a standardized pollinator monitoring scheme[28] will increase the understanding of future changes in pollinator abundance and potential consequences for pollination services. Finally, as with the majority of unstructured UK biological records datasets, there was a southern bias to the data in the study; thus, the trends predominantly reflect changes within this region. However, bees and hoverflies are two of the more well-recorded taxonomic groups in the UK, with an active recorder base and scheme organizers who aim to improve the spatial coverage of data. Given this, and the inclusion of a large number of records from northern Britain, we feel the trends in this study are representative of national-scale trends.

Our full set of model outputs consists of 10,000 samples from the posterior distribution of occupancy for 353 species in each of 34 years (>10$^8$ samples in total). To reduce the computational load of subsequent calculations, we restricted our analysis to a random set of 1000 samples from the posterior of each species:year combination. All trends and other summary statistics were calculated from this set, from which we report median and 95% CIs. Species occupancy time series were clipped, with annual occupancy estimates before the first record and after the final record, dropped from the study. Individual species trends were estimated as the annual growth rate (percent change per year) between the first and final year of the clipped series.

We calculated multispecies composite trends to provide an indicator of the overall trend trajectory for different ecologically significant groups of pollinators

(as seen in Fig. 2 and Supplementary Figures 3, 4, and 5). Occupancy estimates were logged and fed into a linear model with year and species treated as categorical explanatory variables. Sum contrasts were used to ensure the composite trend reflects the average species response. The parameter estimates for the year effects were converted back to the occupancy scale and used as our composite trend metric, effectively a geometric mean occupancy estimate each year across species.

In addition to calculating geometric mean occupancy, we examined temporal patterns in the balance between rare and common species, defined in terms of low and high occupancy, and measured using Simpson's evenness (the $-\log_e D_j$ formulation[26]). Decreases in evenness are indicative of diversity loss and can be considered a signal of biotic homogenization, i.e., communities becoming dominated by a small number of widespread species. Again, using 1000 sampled values from the posterior distribution allowed full propagation of uncertainty. We extracted the first derivatives (i.e., the difference between adjacent years) of geometric mean occupancy and evenness to highlight notable years of change.

**Trait and assemblage classification.** We examined change across five grouping variables aimed at improving our insight into the key drivers of change and potential implications for pollination services. First, we divided species into their broad taxonomic group (splitting bees and hoverflies). This reflects fundamental differences in breeding ecology, with bees being fixed-place foragers that must provision a nest. Next, with a particular focus on implications for pollination services, we examined composite trends for bee species known to be dominant crop pollinators[33] compared with those of other wild bee species. We used CLUS-TASPEC[34] to split species into four categories based on their distribution patterns at the 10 km$^2$ grid square scale, resulting in the following four categories, upland species, southern species, widespread southern species, and widespread species (predominantly hoverflies). To aid visualization of these species clusters, richness maps (using 10 km records between 1980 and 2013) of the clusters are shown in Supplementary Figure 6. Finally, evidence from previous studies suggests that sociality can affect species' sensitivity to environmental change through links to reproductive and foraging capacity[35]. Eusocial species are functionally distinct from other bee species[24] and many are economically important pollinators (7 species were included in the 22 species of dominant crop pollinator). As a group, they have been actively targeted by conservation measures, including the planting of legumes in flower-rich field margins as part of agri-environment schemes. However, the increased foraging capacity of social species may lead to increased pesticide exposure[6] compared with solitary species. We therefore compare composite metrics separately for eusocial and solitary species. A detailed breakdown of which species were in each category can be found in Supplementary Data 1.

**Code availability.** The code used to produce the results and figures in this paper is available from the corresponding author upon request. The occupancy models in this study were run using R2jags[32] via the occDetFunc function, which is freely available as part of the R package Sparta[36].

## Data availability
Data are available from the corresponding author upon reasonable request.

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

## Acknowledgements
We thank Charlie Outhwaite, David Roy, Colin Harrower, James Bullock, Rowan Edwards, and Stuart Roberts for insightful discussion of this manuscript. In addition, we thank the committee and members of BWARS (the UK Bees, Wasps and Ants Recording Society) and the UK Hoverfly Recording Scheme for access to their data, and we are indebted to the volunteers who contribute species' records to these two recording schemes. This study was funded by the UK Joint Nature Conservation Committee, the Natural Environment Research Council (through National Capability funding), by Defra, and the Scottish Government under project WC1101, and by Defra, JNCC, the Welsh

Government, Scottish Government, and partners of the UK Pollinator Monitoring and Research Partnership under project BE0125. The research was partly funded by Natural Environment Research Council and the Biotechnology and Biological Sciences Research Council (BBSRC) under research programmes NE/N018125/1LTS-M ASSIST–Achieving Sustainable Agricultural Systems, and by the Natural Environment Research Council award number NE/R016429/1 as part of the UK-SCAPE programme delivering National Capability.

## Author contributions

G.D.P., C.C., M.E., R.K.A.M., H.E.R., B.A.W., and N.J.B.I. designed the study and contributed to the writing of the manuscript. G.D.P. performed the analysis and led the writing of the manuscript.

## Additional information

**Competing interests:** The authors declare no competing interests.

