## [Peer Review File · Nature Communications]

Reviewers' comments:

Reviewer #1 (Remarks to the Author):

Pollinator declines have been much-discussed in recent years, but hard data are often lacking. There are no long-term data on wild bee or hoverfly populations, but the UK does have a uniquely detailed data set on the distributions of these insects and how they have changed over time. This study takes advantage of this to quantify changes in range of 353 wild bee and hoverfly species. There are problems in interpreting the data, since the individual records used here were collected in a haphazard and unstructured way, and changing numbers of recorders, and changes in the knowledge, experience and behaviour of recorders are hard to account for. Nonetheless the findings are broadly in line with other recent evidence that pollinators and other insects are still declining. I think that this is an important piece of work.

Specific points:

L28 suggest delete comma

L36 "such metrics are insufficiently sensitive to identify pollination deficits" – the same could be said of the approach used here.

L61 How many grid cells are there in GB?

L131 Insert "of"

L133 & onwards: I should first stress that I am no expert in the modelling approaches used here, so my apologies if I am misunderstanding the approach. I can understand that this approach might account for recorder effort, but can it account for changing skills or interests of recorders? For example, several new and good field guides have been published in recent years. Interest in bumblebees in particular has grown, and Bumblebee Conservation Trust have been training volunteers in identifying some of the more tricky species. Increasing ID skills will increase records for rarer and more cryptic species (such as *Bombus humilis*, used in figure 1 – has it really increased, or are more people looking for it and able to identify it?). Essentially, detectability per species may change over time. To give an example, in the 1990s almost nobody could reliably identify *Bombus ruderalis*. Once characters were discovered that enabled them to be distinguished from the much more common *Bombus hortorum*, records of *ruderalis* started popping up all over the place, and this species appears to have greatly increased. We have no idea if it actually increased. To give another example, growing interest in conserving *Bombus distinguendus* led to people searching hard for it in places where it had not previously been recorded, with the explicit aim of filling in 'empty' grid squares. I would like to see a little more space devoted to consideration of these issues and how they might influence the observed patterns.

Dave Goulson

Reviewer #2 (Remarks to the Author):

The authors compiled a huge database on British pollinators in order to study their temporal change. The ms is easy to follow and well-focused. Please, find comments below, which can help in improving it.

Major comments:

1. L56: I'd be most interested in recent changes, i.e. after 2013. There was already a CAP reform since then, etc. Why these data were not included or even excluded (L119)?

2. L128-133: Nice that the authors could control for temporal bias due to greater numbers of recent records. What about potential spatial bias? As far as I know in monitoring programmes there is often a spatial aggregation of records closer to large cities, where most volunteers are available. Is this also reflected in Fig. S6d? Would it be possible to control for this?

Minor comments:

1. L14: This absolute number of occupied cells is not very informative for me (without knowing from how many). I recommend presenting this result only as % change like later on (cf. L18). See also L61-2.
2. L47-8: please, add info how large share is this 353 species out of overall British pollinator species pool.
3. L70: change "all the declines" to "all severe declines".
4. L83 and elsewhere: I recommend calling non-eusocial bees as solitary bees.

Hope this helps to improve the ms,
Péter Batáry

Reviewer #3 (Remarks to the Author):

Insect populations are thought to be declining. Insect declines are a problem because they could affect pollination, a crucial ecosystem service. Unfortunately, there is not much data to show large-scale and long-term declines. Some of the best data is very local (e.g., Vogel 2017 *Science* 356: 576-579). Gary Powney et al. use an occupancy model to derive species-specific trends in occupancy across the UK. They show that many species of bees and hoverflies are indeed in decline. Interestingly, while most species declined, crop pollinators increased. Does this result suggest that ecosystem services (pollination) are not a sufficient reason to protect biodiversity?

The analysis is based on 1 km² grid cells (line 50). Why was this size chosen? Spatial scale will determine the outcome of the analysis. 1 km² grid cells are convenient but this is not a scale determined by biology. Fithian & Hastie 2013 *Annals of Applied Statistics* 7: 1917-1939 have an interesting section on how scale affects the interpretation of occupancy models. It is also worth noting that Redhead et al. *Ecology Letters* (in press) use essentially the same biological records data but construct networks at a different scale (10 x 10 km). Some occupancy estimates shown in the figure are close to zero, i.e. the edge of the parameter space. Using a different scale (2x2 or 5x5 km) could lead to larger occupancy estimates and perhaps more robust trends.

When you use occupancy, you need to define what 100% is. I found it a bit hard to figure out what 100% for the different species is. Is it 12'849 and 12'076 km² for hoverflies and bees, respectively?

I liked the fact that you used occupancy as the metric to describe change (see also van Strien et al. 2011 *Ecological Applications* 21: 2510-2520 and Cruickshank et al. 2016 *Conservation Biology* 30: 1112-1121). I think it is much better than species richness. Can you calculate species richness and show that occupancy is a really a better metric to describe declines? To calculate species richness, you only need to sum the grid cell-specific occurrence probabilities of all species (Dorazio & Royle 2005 *Journal of the American Statistical Association* 100: 389-398; and other papers by these authors).

715'393 records are a lot but could you please describe how many records you have per year, species and grid cell? I did some back-of-envelope calculations and was surprised to see how little data lead to remarkably robust estimates of occupancy (when eye-balling the width of the credible intervals). In the methods section, you write that you had 417'856 records for hoverflies and 214 species of hoverflies. So, on average there are 1952 records per species. Given that you have ~30 years of data, there are ~65 records per year (I would guess that the number of records is

increasing through time). This implies that there is a very small number of records per grid cell. Is this correct? If this amount of data is sufficient for the estimation of occupancy trends, then one may use data from other biological records centres to estimate trends in insect occupancy across a large spatial scale (e.g. Europe). This would be important to better understand the phenomenon of "insect decline". I think this is something perhaps worth emphasizing in the discussion. Most people tend to think that there is not enough data to describe insect/pollinator declines but your analysis may suggest otherwise.

The occupancy model is fine but it is not the standard parameterization of dynamic occupancy models. Such models estimate initial occupancy, extinction and colonization. I was wondering whether it would make sense to estimate the parameters of the dynamic occupancy process rather than occupancy itself? After all, you don't seem to care about occupancy but rather about changes in occupancy.

You fit a separate model for every species. Did you consider multi-species occupancy models? With such an approach you might share information across species and you may be able to derive occupancy estimates for the rare species as well (see Guillera-Arroita 2017 *Ecography* 40: 281-295 for references). Such a multi-species (or community) models seems useful since you present "average occupancy" in figure 2 (please explain in the methods section how average occupancy and the associated credible interval was calculated (simply the arithmetic mean?; the legend to figure S2 suggests it is a geometric mean)).

The paper describes trends and shows that some species decline more than others. Is there data which could be used to explain variation in trends (primarily spatial, I think)? You talk about agri-environment schemes, habitat loss, climate change and pesticides? The paper by Redhead et al. *Ecology Letters* (in press) suggests that there are some environmental data. The use of such data might lead to an even more interesting analysis. Miller et al. 2018 *Nature Communications* 9: 3926 is a nice example (methods and results). Last but not least, is there data on crop yield and could you link such data to pollinator occupancy?

Line 14, 61. 2.7 million occupied 1 km² grid cells is a lot but I would need a reference. What is the total number of grid cells in the analysis? Can you express the loss as a proportion of how many grid cells were occupied in the past?

Line 48. I think you should cite Royle and Kéry 2007 *Ecology* 88: 1813-1823 here. These authors describe the Bayesian state-space formulation of dynamic occupancy models.

Line 68. Why do you describe the credible intervals as 30 to 21%? 21-30% would be more usual.

Line 85. Is there data which could be used to link agri-environmental schemes directly to trends? For example, one might use the proportion of set asides within a grid cell as a covariate.

Line 120. You excluded cells with less than two years of data. Above I commented on the amount of data per species and year. What is the average number of years for which you have data (per cell)? This seems important because you clip time series (line 175).

Line 128. I think you should state explicitly that the data are presence-only records based on lists. This implies that you somehow have to create the non-detection data for the occupancy models. Please explain how this was done. A reference to a paper which explains the process may be sufficient (e.g., Kéry et al. 2010 *Journal of Biogeography* 37: 1851-1862 or Kéry et al. 2010 *Conservation Biology* 24: 1388-1397).

Line 141-152. Please provide more information on the priors (e.g. $a[t]$).

Line 142. $b[t]$ is a fixed effect and $u[i]$ is a random effect, right? "fixed" and "random" are more

precise and informative than “categorical”.

Line 144. I think this should be “conditional upon $z[i,t] = 1$ ”.

Line 149. While it is clear how day-lists were used in the model (= equation on line 140), I didn't understand how “1) single species recorded” were included in the model. Please explain.

Line 154. If you state that 2.7 million occupied grid cells were lost, then it seems as if you estimate occupancy probability and additionally the finite sample estimate of occupancy (see Royle and Kery 2007 Ecology 88: 1813-1823)? Is this the case?

Line 155-156. Here you state that you have 3 chains and 20'000 iterations per chain. Burn-in is 10'000. You don't state whether the chains were thinned. Thus, you keep $3 \times 10'000$ iterations. Further below (e.g line 173) you state that you have 1000 samples from the posterior. What happened to the other 29'000 samples? Did I misunderstand something?

Line 155. You may want to replace reference #30 with a reference to JAGS and the R package R2Jags.

Line 155, 216-217. Did you use R2Jags or the Sparta R package? Does occDetFunc call R2Jags?

Line 155. How did you assess convergence?

Line 173. I would not use the term “data set” for samples from the posterior.

Line 176-177. I don't understand how trends were calculated. Simply as $\text{trend} = \text{psi}[\text{last year in time series}] / \text{psi}[\text{first year in time series}]$? This should work but it means that you basically throw away all intermediate estimates of psi.

Line 345, 348. Please explain how you calculated the first derivative of occupancy and evenness.

Line 350. Error bars show the upper and lower limits of the credible interval rather than the lower and upper credible intervals.

Line 424. Why do you use the credible interval? A better way of doing this would be the proportion of the posterior which is greater or smaller than zero. If 95% of the posterior are greater than 0, then this is evidence for a positive trend. If 95% of the posterior are smaller than 0, then this is evidence for a positive trend. If the proportions are not 95%, then there is no clear evidence for a trend. You could also use this approach to compute a probability of decline. For an example, see Buckley et al. 2014 Animal Conservation 17: 27-34 (based on Wade 2000 Conservation Biology 14: 1308-1316).

Reviewer #1:

L28 suggest delete comma

Done

L36 “such metrics are insufficiently sensitive to identify pollination deficits” – the same could be said of the approach used here.

The reviewer makes a good point that occupancy trends may not be sufficient to reveal pollination deficits. The point we’re making here is that occupancy trends are far closer to the level of resolution that would be required to identify such deficits. We have a section (lines 109-112) of the discussion highlighting this point, linking to an increased need for abundance data to better understand deficits in pollination services.

L61 How many grid cells are there in GB?

There are approximately 240,000 1km grid cells in GB. We’ve added this number into the manuscript along with an explanation for how we derived the 2.7 million figure (lines 60-65).

L131 Insert “of”

Done

L133 & onwards: I should first stress that I am no expert in the modelling approaches used here, so my apologies if I am misunderstanding the approach. I can understand that this approach might account for recorder effort, but can it account for changing skills or interests of recorders? For example, several new and good field guides have been published in recent years. Interest in bumblebees in particular has grown, and Bumblebee Conservation Trust have been training volunteers in identifying some of the more tricky species. Increasing ID skills will increase records for rarer and more cryptic species (such as *Bombus humilis*, used in figure 1 – has it really increased, or are more people looking for it and able to identify it?). Essentially, detectability per species may change over time. To give an example, in the 1990s almost nobody could reliably identify *Bombus ruderatus*. Once characters were discovered that enabled them to be distinguished from the much more common *Bombus hortorum*, records of *ruderatus* started popping up all over the place, and this species appears to have greatly increased. We have no idea if it actually increased. To give another example, growing interest in conserving *Bombus distinguendus* led to people searching hard for it in places where it had not previously been recorded, with the explicit aim of filling in ‘empty’ grid squares. I would like to see a little more space devoted to consideration of these issues and how they might influence the observed patterns.

We estimate detectability separately for each species and it is allowed to vary over time (via a categorical year effect in the detection model). This approach should allow the models to account for the impact of changes in recorder behaviour, such as the increased ID skills in response to the publication of a new field guide. Simulations presented in Isaac et al (2014 – cited in the main manuscript) showed the occupancy modelling approach is robust to a range of known biases, including systematic trends in the detectability of individual species, such as might occur following the publication of a new field guide. However, we accept the potential exists for additional or more extreme forms of bias that are not fully accounted for in the detection model, challenging the reliability of our occupancy estimates. These issues are discussed within methods section (lines 174-181), additionally we have added a statement to the methods (lines 151-152), clarifying the role of the year effect in the detection model

Reviewer #2:

Major comments:

1. L56: I'd be most interested in recent changes, i.e. after 2013. There was already a CAP reform since then, etc. Why these data were not included or even excluded (L119)?

There is a significant time lag between the collection of the biological records and their availability for analysis. Specifically, the number of records available for 2014 is just ~25% of the number for 2013, so we excluded data post-2013 knowing these were incomplete datasets.

2. L128-133: Nice that the authors could control for temporal bias due to greater numbers of recent records. What about potential spatial bias? As far as I know in monitoring programmes there is often a spatial aggregation of records closer to large cities, where most volunteers are available. Is this also reflected in Fig. S6d? Would it be possible to control for this?

The reviewer is correct and brings up an important point that we have now discussed in the methods section (lines 184-189). As with the majority of unstructured UK biological records datasets, there was a southern bias to the data in the study. In extreme cases, this may mean national-scale trends predominantly reflect trends within southern Britain. However, bees and hoverflies are two of the more well-recorded taxonomic groups in the UK, with an active recorder base and scheme organisers who aim to improve the spatial coverage of data. Given this, and the inclusion of a large number of records from northern Britain, we feel the trends in this study are representative of national-scale trends.

Minor comments:

1. L14: This absolute number of occupied cells is not very informative for me (without knowing from how many). I recommend presenting this result only as % change like later on (cf. L18). See also L61- We thank the reviewer for highlighting the lack of clarity surrounding this figure. We've now added a clear explanation for how this figure was derived with reference to the total number of 1km grid cells in GB. (lines 60-65).

2. L47-8: please, add info how larger share is this 353 species out of overall British pollinator species pool.

There is no clear definition by which it's possible to identify the complete set of pollinator species. For example, butterflies, flies (other than hoverflies) and some beetles visit flowers, but their contribution to pollination services is not known. It's uncontroversial that the majority of bees and hoverflies are pollinators, with the exception of a few social parasites (which we excluded). The total number of pollinators among bee and hoverfly species is about 470 (depending on whether some occasionally recorded species are considered as vagrants or native), so our models cover 75% of the species in these groups. We now quote this figure explicitly on line 132.

3. L70: change "all the declines" to "all severe declines".

Done

4. L83 and elsewhere: I recommend calling non-eusocial bees as solitary bees.

Done

Reviewer #3:

Interestingly, while most species declined, crop pollinators increased. Does this result suggest that ecosystem services (pollination) are not a sufficient reason to protect biodiversity?

This is an interesting point and one much debated in the pollinator literature (please see Kleijn et al., Nature Comms. 6, 7414, 2015). The counter-arguments are 1) studies have shown yields of pollinator dependent crops are dependent on the diversity and total abundance of pollinators, and 2) the pollination of wild flowers can still be considered a valuable ecosystem service, with flowers being particularly important in terms of culture values. However, as the aim of our study is to report

trends in wild pollinators, we feel it is outside the scope of this study to enter this debate, other than highlighting the potential value of conservation efforts aimed at protected pollinator species (lines 20-23).

The analysis is based on 1 km² grid cells (line 50). Why was this size chosen? Spatial scale will determine the outcome of the analysis. 1 km² grid cells are convenient but this is not a scale determined by biology. Fithian & Hastie 2013 *Annals of Applied Statistics* 7: 1917-1939 have an interesting section on how scale affects the interpretation of occupancy models. It is also worth noting that Redhead et al. *Ecology Letters* (in press) use essentially the same biological records data but construct networks at a different scale (10 x 10 km). Some occupancy estimates shown in the figure are close to zero, i.e. the edge of the parameter space. Using a different scale (2x2 or 5x5 km) could lead to larger occupancy estimates and perhaps more robust trends.

The point about scale dependency (Fithian & Hastie) is well made. It is a constraint of any model based in discrete space (all occupancy models and the vast majority of SDMs) that a choice of scale must be made. Given this necessity, the most appropriate choice reflects the assumptions and statistical power of the test being deployed. Occupancy models were conceived to estimate parameters of biological populations. Thus, the most appropriate choice is the scale that best reflects the characteristic scale at which organisms use the environment. Given that bees and hoverflies are mobile organisms, we feel that 1km grid cell is far more appropriate than either 10km grid or 100m (i.e. 1 hectare grid cells). Larger grids would undoubtedly include multiple populations of the same species, thus greatly reducing our power to detect biologically-meaningful change; smaller grid cells (e.g. a hectare) would increase the probability that individual bee and hoverfly populations span multiple grid cells, thus violating the assumption that cells are mutually independent. One could argue about the relative merits of grid cells of 1km vs 2km or some other dimension, but there is really no objective way to resolve the issue. Reassuringly, preliminary analyses looking into the impact of changing the spatial scale of modelled data showed little impact on the trend estimates at 1km and 10km (manuscript in prep). In response to the reviewer's point, we have added a sentence (line 127-128) justifying the choice of grid cell.

When you use occupancy, you need to define what 100% is. I found it a bit hard to figure out what 100% for the different species is. Is it 12'849 and 12'076 km² for hoverflies and bees, respectively? The reviewer's interpretation is correct, and we agree it is an important point of clarification that we missed out in our initial submission. This has now been added (lines 161-163)

I liked the fact that you used occupancy as the metric to describe change (see also van Strien et al. 2011 *Ecological Applications* 21: 2510-2520 and Cruickshank et al. 2016 *Conservation Biology* 30: 1112-1121). I think it is much better than species richness. Can you calculate species richness and show that occupancy is a really a better metric to describe declines? To calculate species richness, you only need to sum the grid cell-specific occurrence probabilities of all species (Dorazio & Royle 2005 *Journal of the American Statistical Association* 100: 389-398; and other papers by these authors).

The reviewer makes an interesting suggestion. Conceptually, species richness and occupancy are tightly linked: the extinction of one species from one site reduces both occupancy for that species and richness for that site, and thus the aggregates for both statistics. We have done some simulations to explore this: the richness change and occupancy change are tightly correlated, as one might expect, but the slope of the relationship depends on 1) the frequency distribution of occupancy across species, 2) the correlation in species' distributions across sites and 3) whether (or not) species are lost at random. These simulations are part of another manuscript that is currently in review, so we don't feel it is appropriate to report them in this manuscript. We also feel that to raise the issue of this topic would distract from the main message of our study. Furthermore, any discussion of the topic would be complicated by the scale dependency issue discussed above –

previous studies of bee and hoverfly declines studied change in species richness at 10km grid cells or coarser, and it is well-established that species richness trends are highly sensitive to spatial scale (Keil et al 2011 *Ecography* 34: 392-401) – arguably more sensitive than occupancy changes.

715'393 records are a lot but could you please describe how many records you have per year, species and grid cell? I did some back-of-envelope calculations and was surprised to see how little data lead to remarkably robust estimates of occupancy (when eye-balling the width of the credible intervals). In the methods section, you write that you had 417'856 records for hoverflies and 214 species of hoverflies. So, on average there are 1952 records per species. Given that you have ~30 years of data, there are ~65 records per year (I would guess that the number of records is increasing through time). This implies that there is a very small number of records per grid cell. Is this correct? If this amount of data is sufficient for the estimation of occupancy trends, then one may use data from other biological records centres to estimate trends in insect occupancy across a large spatial scale (e.g. Europe). This would be important to better understand the phenomenon of “insect decline”. I think this is something perhaps worth emphasizing in the discussion. Most people tend to think that there is not enough data to describe insect/pollinator declines but your analysis may suggest otherwise.

Species vary in the number of records included in the analysis. The reviewer is correct, there are species where trend estimates are derived from a low number of records. As the main focus of the study is trends in pollinators, we feel a discussion of the value of occupancy models is beyond the scope of this manuscript. Furthermore, a paper (Outhwaite et al 2018; ref 31 in the manuscript) solely aimed at describing the value of occupancy models for modelling rare species has recently been published. We cite this study in the methods section on multiple occasions to draw attention to the potential value of occupancy models.

The occupancy model is fine but it is not the standard parameterization of dynamic occupancy models. Such models estimate initial occupancy, extinction and colonization. I was wondering whether it would make sense to estimate the parameters of the dynamic occupancy process rather than occupancy itself? After all, you don't seem to care about occupancy but rather about changes in occupancy.

Dynamic models require estimation of both extinction and colonization rates for each year, whereas our static model estimates only occupancy, i.e. the dynamic model estimates double the number of parameters. The random walk formulation (Outhwaite et al 2018; ref 31 in the manuscript) allows us to share information across years (as in the dynamic model) whilst estimating fewer parameters. These issues are explored in more detail in Outhwaite et al 2018, who also showed that the random walk model produces reliable occupancy trends when the data are sparse (i.e. number of records per species per year <50). Our experience with dynamic models is that the data available are insufficient to estimate both colonization and extinction for the vast majority of species (the posterior distribution for many years span the full range of possible values), unless multiple parameters are shared across species (see next point).

You fit a separate model for every species. Did you consider multi-species occupancy models? With such an approach you might share information across species and you may be able to derive occupancy estimates for the rare species as well (see Guillera-Arroita 2017 *Ecography* 40: 281-295 for references). Such a multi-species (or community) models seems useful since you present “average occupancy” in figure 2 (please explain in the methods section how average occupancy and the associated credible interval was calculated (simply the arithmetic mean?; the legend to figure S2 suggests it is a geometric mean)).

We agree with the reviewer that there are valid reasons for using multi-species occupancy models, particularly the ability to share information across species to improve occupancy estimates for rare species. However, an issue with multi-species occupancy models is they assume (to some extent)

species share an underlying response to environmental change. This can be an unhelpful assumption when looking across a large number of species with a desire to produce species-specific trends as we do in this paper: specifically, the trends for species with few data would be driven mostly by the mean of other species.

The description of how we calculated average occupancy is covered in lines 198 to 204 of the methods section. We've added a link to the figures into the methods section (line 199) to improve clarity.

The paper describes trends and shows that some species decline more than others. Is there data which could be used to explain variation in trends (primarily spatial, I think)? You talk about agri-environment schemes, habitat loss, climate change and pesticides? The paper by Redhead et al. Ecology Letters (in press) suggests that there are some environmental data. The use of such data might lead to an even more interesting analysis. Miller et al. 2018 Nature Communications 9: 3926 is a nice example (methods and results). Last but not least, is there data on crop yield and could you link such data to pollinator occupancy?

We did consider tests to examine the key drivers of the variation in trends between species, but felt that this was beyond the scope of this paper. Essentially, we feel the core aim of this paper was to produce species-specific pollinator trends, filling this gap in the evidence base and recent changes in the status of pollinators in Britain. To introduce data on potential drivers would change this focus, and raise complicated questions about whether the drivers selected truly capture the full range of threats facing wild pollinators. Studying drivers is further complicated by the nature of the data that are available: 1) the habitat data are compromised by incompatibilities between different land cover maps, and that land cover is an imperfect measure of land-use and the intensity thereof; 2) the agricultural data (on yield, pesticides and agri-environment schemes) contains gaps in space and time that require models and/or assumptions to fill. We feel these issues, whilst important, would require a dedicated study in order to be addressed with due rigour, and therefore fall outside the scope of the current manuscript

Line 14, 61. 2.7 million occupied 1 km² grid cells is a lot but I would need a reference. What is the total number of grid cells in the analysis? Can you express the loss as a proportion of how many grid cells were occupied in the past?

The reviewer is correct that we don't adequately explain where this figure of 2.7 million grid cells derives from. The calculation is based upon 1) converting the change in occupancy from a proportion into a number of grid cells gained and lost for each species, 2) extrapolating from the grid cells in our study (12,076 for bees and 12,849 for hoverflies, as reported on line 126) to the full set of ~240,000 grid cells in Great Britain. 3) Summing across species. Step 2 makes the simplifying assumption that the grid cells in our data are a random sample of the cells in GB. Thus, the number we quote is simply another way of expressing the change in the proportion of occupied grid cells. We've an explanation for how we derived the 2.7 million figure (lines 60-65).

Line 48. I think you should cite Royle and Kery 2007 Ecology 88: 1813-1823 here. These authors describe the Bayesian state-space formulation of dynamic occupancy models.

We have now included this citation into the manuscript

Line 68. Why do you describe the credible intervals as 30 to 21%? 21-30% would be more usual.

We agree 21-30% is more usual, and we have now corrected this throughout the manuscript.

Originally these confidence intervals were preceded with the direction of the trend (e.g. -30 to -21%)

Line 85. Is there data which could be used to link agri-environmental schemes directly to trends? For example, one might use the proportion of set asides within a grid cell as a covariate.

We agree that directly linking environmental factors to the observed trends would be interesting. However, as discussed above, we feel this is beyond the scope of this study.

Line 120. You excluded cells with less than two years of data. Above I commented on the amount of data per species and year. What is the average number of years for which you have data (per cell)? This seems important because you clip time series (line 175).

The average number of years with bee records per site in the analysis is 3.8. For hoverflies the number is 3.7. The structure of our model, means that sites with few years contribute little information to the overall trend, because the occupancy at such sites is estimated very imprecisely. We include them because Kamp et al (2016, doi:10.1111/ddi.12463) showed that power of such models declines substantially when data-poor sites are excluded. We exclude sites with single years since these contribute no signal and only noise to the overall trend (as shown in Isaac et al 2014).

Line 128. I think you should state explicitly that the data are presence-only records based on lists. This implies that you somehow have to create the non-detection data for the occupancy models. Please explain how this was done. A reference to a paper which explains the process may be sufficient (e.g., Kéry et al. 2010 Journal of Biogeography 37: 1851-1862 or Kéry et al. 2010 Conservation Biology 24: 1388-1397).

We now explicitly state that these models are based on presence-only data with inferred non-detections (lines 124-126)

Line 141-152. Please provide more information on the priors (e.g. $a[t]$).

We have now added to the description of the priors in the methods section. We state that we use the random walk half-cauchy prior formulations of the occupancy model of Outhwaite et al. (2018), with the remaining parameters given uninformative priors.

Line 142. $b[t]$ is a fixed effect and $u[i]$ is a random effect, right? “fixed” and “random” are more precise and informative than “categorical”.

This is correct and we’ve updated the manuscript accordingly

Line 144. I think this should be “conditional upon $z[i,t] = 1$ ”.

Technically yes, it would be correct to state that detection is conditional on z taking a value of 1 (the site is truly occupied). However, since z is defined as binary (occupied or not), we feel the extra detail is superfluous.

Line 149. While it is clear how day-lists were used in the model (= equation on line 140), I didn’t understand how “1) single species recorded” were included in the model. Please explain.

We have replaced “recorded” with “lists” to clarify that a single species record is also a list of one.

Line 154. If you state that 2.7 million occupied grid cells were lost, then it seems as if you estimate occupancy probability and additionally the finite sample estimate of occupancy (see Royle and Kery 2007 Ecology 88: 1813-1823)? Is this the case?

Yes, we did estimate both quantities. As stated above, the 2.7 million figure is an extrapolation from the results across species.

Line 155-156. Here you state that you have 3 chains and 20’000 iterations per chain. Burn-in is 10’000. You don’t state whether the chains were thinned. Thus, you keep $3 \times 10’000$ iterations. Further below (e.g line 173) you state that you have 1000 samples from the posterior. What happened to the other 29’000 samples? Did I misunderstand something?

We have reworked this section clarifying the reason why we base our summary trends on 1000 samples from the posterior (lines 190-197). Additionally, we have added the thinning rate (3) used to the manuscript.

Line 155. You may want to replace reference #30 with a reference to JAGS and the R package R2Jags. We thank the reviewer for pointing this out. We have corrected this in the text (lines xxx).

Line 155, 216-217. Did you use R2Jags or the Sparta R package? Does occDetFunc call R2Jags? We used the occDetFunc function from Sparta that calls R2Jags, we have clarified this in the text (line 167 and in the 'code availability' section).

Line 155. How did you assess convergence?

We have added a sentence (now at the end of this paragraph) "This was sufficient to achieve convergence ($R_{hat} < 1.1$) for the vast majority of occupancy estimates across species and years: we retained the small number of combinations for which $R_{hat} > 1.1$ since they are unlikely to exert directional bias on our high-level summary statistics."

Line 173. I would not use the term "data set" for samples from the posterior.

We have reworked this section and removed the term "dataset", as requested (lines 190-197).

Line 176-177. I don't understand how trends were calculated. Simply as $trend = \psi_i[\text{last year in time series}] / \psi_i[\text{first year in time series}]$? This should work but it means that you basically throw away all intermediate estimates of ψ_i .

The reviewer is correct that our species-specific trend estimates do not use the intermediate values of ψ_i . This is because the intermediate values are not required: the annual growth rate between first and last years is simply the sum of all the annual growth rates between adjacent years. We feel justified in using mean growth rate as our trend measure because most species' trends were smooth as opposed to chaotic (e.g. figure 1). Note that intermediate values would have been required to estimate linear trends for each species, but this seemed unreasonable given that many species are demonstrably nonlinear (e.g. *Bombus humilis* in figure 1).

Line 345, 348. Please explain how you calculated the first derivative of occupancy and evenness. This has been clarified (lines 210-212).

Line 350. Error bars show the upper and lower limits of the credible interval rather than the lower and upper credible intervals.

Thanks for pointing out this error, which has been corrected on all figure legends.

Line 424. Why do you use the credible interval? A better way of doing this would be the proportion of the posterior which is greater or smaller than zero. If 95% of the posterior are greater than 0, then this is evidence for a positive trend. If 95% of the posterior are smaller than 0, then this is evidence for a positive trend. If the proportions are not 95%, then there is no clear evidence for a trend. You could also use this approach to compute a probability of decline. For an example, see Buckley et al. 2014 Animal Conservation 17: 27-34 (based on Wade 2000 Conservation Biology 14: 1308-1316). The reviewer's point is valid, if the goal is to test the null hypothesis of no change. However, this was not our goal.

REVIEWERS' COMMENTS:

Reviewer #1 (Remarks to the Author):

I am content that the authors have responded adequately to the points raised by myself and the other reviewers.

Dave Goulson

Reviewer #3 (Remarks to the Author):

The revised manuscript is much better than the previous draft. The "response to reviewers" letter clearly explains how the manuscript was revised. The authors revised the manuscript in response to my comments or they gave convincing reasons why they did not do so.

Nevertheless, I have some additional comments.

Line 57. "calculated as the annual growth rate in occupancy". The authors explained that they calculated trend as $\text{occupancy}[\text{last year}]/\text{occupancy}[\text{first year}]$. Thus, the explanation how species-level trends were calculated should be changed. "Annual growth rates" is a bit misleading.

Line 14, 222. Should this be "1km² [or 1x1 km] grid cells" rather than "1 km grid cell"? & Should this be "100 km² [or 10x10 km] records" rather than "10 km records"?)?

Line 79. Typo in "cared" ("ca'red" in my copy of the manuscript).

Line 151. This should really be "upon $z[i,t] = 1$ ". I don't think that this a superfluous extra detail. Detectability is not conditional on z , it is conditional on $z=1$. Detectability can't be estimated if the species is absent ($z=0$).

Line 172-189. This paragraph was revised in response to comments made by another reviewer. While I agree, I think there is an important point missing. The detection part of the occupancy model may not be perfect but it is certainly better than a model which ignores detectability. By the way, visual inspection of figure 1 suggests that increasing numbers of records don't lead to much better estimates. For example, the credible interval for the blue species (*Bombus humilis*) is not much narrower in recent years than in the 1980ties.

REVIEWERS' COMMENTS:

Reviewer #3 (Remarks to the Author):

Line 57. “calculated as the annual growth rate in occupancy”. The authors explained that they calculated trend as $\text{occupancy}[\text{last year}]/\text{occupancy}[\text{first year}]$. Thus, the explanation how species-level trends were calculated should be changed. “Annual growth rates” is a bit misleading.

We have added clarity by stating that the annual growth rate was calculated as the percent change per year between the first and last year (Lines 79-80 and in the methods section on Line 228)

Line 14, 222. Should this be “1km² [or 1x1 km] grid cells” rather than “1 km grid cell”? & Should this be “100 km² [or 10x10 km] records” rather than “10 km records”)?

The reviewer is correct, we’ve updated the manuscript accordingly (Lines 28 and 253)

Line 79. Typo in “cared” (“ca’red” in my copy of the manuscript).

We thank the reviewer for spotting this. This has been corrected (Line 79)

Line 151. This should really be “upon $z[i,t] = 1$ ”. I don’t think that this a superfluous extra detail. Detectability is not conditional on z , it is conditional on $z=1$. Detectability can’t be estimated if the species is absent ($z=0$).

We have added this into the manuscript (Line 180)

Line 172-189. This paragraph was revised in response to comments made by another reviewer. While I agree, I think there is an important point missing. The detection part of the occupancy model may not be perfect but it is certainly better than a model which ignores detectability. By the way, visual inspection of figure 1 suggests that increasing numbers of records don’t lead to much better estimates. For example, the credible interval for the blue species (*Bombus humilis*) is not much narrower in recent years than in the 1980ties.

We agree with the reviewer that it is worth stating that, while not perfect for all species, the model will almost certainly be better than a model that ignores variation in detectability (please see the additions to lines 210-211). The uncertainty in *Bombus humilis* occupancy estimates are fairly consistent over time, however, this is not the case for all species. The uncertainty for most species resembles the pattern seen in the other example species (*Colletes succinctus*).